# Determining Medication Errors in an Adult Intensive Care Unit

**DOI:** 10.3390/ijerph20186788

**Published:** 2023-09-20

**Authors:** Renata da Nóbrega Souza de Castro, Lucas Barbosa de Aguiar, Cris Renata Grou Volpe, Calliandra Maria de Souza Silva, Izabel Cristina Rodrigues da Silva, Marina Morato Stival, Everton Nunes da Silva, Micheline Marie Milward de Azevedo Meiners, Silvana Schwerz Funghetto

**Affiliations:** 1Graduate Program in Health Sciences and Technologies, Faculty of Ceilandia, University of Brasília, Federal District, Brasília 72220-275, Brazil; renatadanobrega@gmail.com (R.d.N.S.d.C.); kucslucs@hotmail.com (L.B.d.A.); cdssilva@gmail.com (C.M.d.S.S.); marinamorato@unb.br (M.M.S.); evertonsilva@unb.br (E.N.d.S.); silvanasf@unb.br (S.S.F.); 2Department of Nursing, Faculty of Ceilandia, University of Brasília, Federal District, Brasília 72220-275, Brazil; crgrou@unb.br; 3Department of Pharmacy, Faculty of Ceilandia, University of Brasília, Federal District, Brasília 72220-275, Brazil; mmmeiners@unb.br

**Keywords:** medication error, patient safety, hospital costs, cost, cost analysis

## Abstract

Introduction: Research addressing the costs of Medication errors (MEs) is still scarce despite issues related to patient safety having significant economic and health impacts, making it imperative to analyze the costs and adverse events related to MEs for a better patient, professional, and institutional safety. Aim: To identify the number of medication errors and verify whether this number was associated with increased hospitalization costs for patients in an Intensive Care Unit (ICU). Method: This retrospective cross-sectional cohort study evaluated secondary data from patients’ electronic medical records to compile variables, create a model, and survey hospitalization costs. The statistical analysis included calculating medication error rates, descriptive analysis, and simple and multivariate regression. Results: The omission error rate showed the highest number of errors per drug dose (59.8%) and total errors observed in the sample (55.31%), followed by the time error rate (26.97%; 24.95%). The omission error had the highest average when analyzing the entire hospitalization (170.40) and day of hospitalization (13.79). Hospitalization costs were significantly and positively correlated with scheduling errors, with an increase of BRL 121.92 (about USD $25.00) (95% CI 43.09; 200.74), and to prescription errors, with an increase of BRL 63.51 (about USD $3.00) (95% CI 29.93; 97.09). Conclusion: We observed an association between two types of medication errors and increased hospitalization costs in an adult ICU (scheduling and prescription errors).

## 1. Introduction

The potential of medication errors (MEs) to cause harm to patients and increase hospital and legal costs is a current concern for the global health system, as reflected in the World Health Organization’s third global challenge [1]. MEs are preventable events prone to occur at all stages of the medication process and are relatively frequent among hospitalized adult patients, with reported rates ranging from 5% to around 90% [2,3,4,5,6].

Among all hospital departments, Intensive Care Units (ICUs) are the hospital environment with perhaps the highest volume of prescribed drugs, which favors errors that can compromise patient safety, increase patient length of stay, and, consequently, increase hospital expenses [7,8]. ME prevalence in ICUs varies and has been documented in the literature for a long time [3,4,9,10,11,12].

Research addressing the costs of ME is still scarce despite issues related to patient safety having significant economic and health impacts. Analyzing the existing literature for ME-related costs and the adverse events related to the medication system’s prescription, dispensing, and administration steps is imperative for better patient, professional, and institutional safety [13].

MEs are estimated to contribute to a total of 12,000 deaths a year in the British National Health Service, which could contribute an additional EUR 0.75 to EUR 1.5 billion to healthcare costs. Worldwide estimates point to an annual expenditure of around USD $42 billion on ME, corresponding to 0.7% of total health expenditures [2].

ME-related issues represent a comparability challenge due to the diversity of contexts, conducts, clinical protocols, and data available from health institutions [13,14,15,16,17]. This gap needs to be addressed by researchers and professionals, preferably in studies that include the stages of the medication chain process on medication errors and their consequences, which impact the patient/family, health professional and institution, knowledge about costs, and the pharmaceutical’s efficacy [13].

Thus, it is paramount to analyze ME-related costs in the existing literature to reduce risks and optimize resources that will hopefully contribute positively to the safety of patients, professionals, and institutions. This study aimed to identify the number of medication errors and verify whether these errors were associated with increased hospitalization costs while controlling for patient characteristics and length of stay in the adult ICU of a hospital from the Federal District, Brazil.

## 2. Methods

### 2.1. Study Type

This retrospective cross-sectional cohort study evaluated secondary data from patients’ electronic medical records to verify the number of medication errors (MEs) and if these were associated with an increase in the included patients’ hospitalization costs. For this, the study was divided into two methodological stages. The first identified MEs and the prevalence of these errors among patients admitted to an adult Intensive Care Unit (ICU), and the second analyzed associated factors by inserting the number of MEs in a linear regression model with one of the hospitalization costs’ explanatory variables while controlling for patient characteristics and length of hospitalization.

### 2.2. Study Sample

The sample consisted of patients admitted to the adult ICU for more than 24 h between August 2018 and February 2019. The sample size was 38 patients with a 10% margin of error, based on a total of 62 patients hospitalized during the period and considering a 5% sampling error and a 95% confidence level.

### 2.3. Location

The research was conducted in a public hospital institution that is part of the Unified Health System (SUS) located in the Federal District (DF). This hospital was specifically selected because it is a training ground (provides internships) that prepares health professionals with ties to public/private universities and has a computerized registration system. This hospital, accredited in medium and high complexity services with outpatient and hospital care, is located in the Federal District’s largest and most populous administrative region, with roughly 500,000 inhabitants, corresponding to the 13th most populated municipality in the country. The selected adult ICU has ten beds, one of which is surgical, with a 14-day-stay mean and a 95% occupation rate. The factors that endorsed this unit selection were the medication volume, the complexity of the medication process (calculation of doses, diversity of injectable medications, and variety of infusions), the users’ clinical conditions, and the types of medication used.

### 2.4. Selection Criteria

Patients over 18 years old with medical records that included identification information and clinical evolution during their period of hospitalization within the unit, which would allow for their evaluation, were included. Twelve patients were excluded: three were excluded due to incomplete medical records, three more were excluded for being under the age of 18, and the remaining six were excluded for staying less than 24 hours in the unit.

### 2.5. Data Collection

Firstly, we identified the patients hospitalized during the study period by perusing the ICU admission book, and their medical records were located on and accessed using the institution’s electronic system. Based on this access, clinical summaries were analyzed to extract demographical, epidemiological, and clinical data. The medication information from all medical prescriptions referring to the patient’s hospitalization period and the nursing record of medications’ scheduling and administration were also collected.

The variables collected were as follows: (a) those related to the patient, including date and time of admission, institutional registration number, date of birth, age, biological sex, origin, diagnosis, and primary International Classification of Diseases (ICD), days of hospitalization, clinical outcome; (b) those related to the treatment, including prescribed medications (therapeutic classes, route of administration, prescribed dose, posology, infusion rate, duration of use, scheduling, and omission).

Seven types of errors described by the National Coordinating Council for Medication Error Reporting and Prevention (NCC MERP) [18] qualified for evaluation and inclusion in this study: dosage, scheduling, route, omission, dispensation, timetable, and prescription. These were measured by previously trained evaluators and analyzed according to the adopted NCC MERP criteria [18].

Dispensing error (error type 5) was defined as an instance where the prescribed medication was incorrectly distributed; omission error (error type 4), when the medication was not administered or was administered but there is no record of it; time error (error type 6), when the medication was administered outside the time interval established by the institution; dosage error (error type 1), when a higher or lower dose was administrated than that recommended for the prescribed medication; administration route error (error type 3), when there was an incompatibility between the prescribed route and the one established in the literature; prescription error (error type 7), when there was at least one nonconformity in the description, the wording of the information, or both on the medication’s correct name, use of abbreviations, dose, frequency, administration route as well as wrong pharmaceutical form, inappropriate indication, duplicate or redundant therapy, documented allergy to prescribed medications, contraindicated therapy, and absence of critical information necessary for the medication dispensing and administration; and scheduling error (error type 2), when there was a lack of correct documentation, incorrect choice of times and intervals, or both for administering medication doses [19,20,21].

Data referring to patients’ hospitalization costs included the direct costs recorded in the hospital admission authorization (HAA), such as expenses related to laboratory and imaging tests, medications, surgical procedures, ICU daily rates, and medical and specialized procedures such as physiotherapeutic care and phonoaudiology (speech and hearing) therapy. Costs refer to the amounts reimbursed by the Ministry of Health to health providers based on the Unified Health System (SUS) database management system (SIGTAP) for procedures, medications, orthoses, prostheses, and special materials. The hospital’s Cost Management Center (CMC) supplied these costs upon our request, which provided the total amount of hospitalization costs approved for reimbursement.

### 2.6. Data Processing and Analysis

The error rate was calculated according to the methodology adopted by PROQUALIS, which corresponds to the number of medications administered with error divided by the total number of medications administered times 100. The total number of medications administered corresponds to all medications prescribed in a given timeframe [22].

For the descriptive analysis, the central tendency (mean), dispersion (standard deviation), and the maximum and minimum amounts per medication error type were calculated. These parameters were calculated using (i) the number of hospitalizations and (ii) the number of days of hospitalization for each medication error.

Seven regression models were created for our regression model analysis, one for each type of error. These models evaluated whether medication errors were associated with increased hospitalization costs. Each regression model ran with two specifications: simple linear model (without controls) and multivariate (with controls). In the simple linear model, the dependent variable was the total hospitalization costs, and the independent variable was the type of error under investigation. In the multivariate model, other independent variables, such as the following, were added to the model: (i) days of hospitalization; (ii) patient’s age at admission; (iii) patient’s biological sex; (iv) whether the patient had hypertension (high blood pressure); (v) whether the patient had diabetes mellitus; (vi) whether the patient was an alcoholic at admission; and (vii) whether the patient was a smoker at admission. These independent variables were chosen as they are known to impact the length of the patient's hospitalization stay.

### 2.7. Ethical Statements

This project complies with Resolution 466/12 of the National Health Council and guarantees the anonymity and confidentiality of the information collected in the medical records and prescriptions from both professionals and users. The University of Brasília’s Ceilândia Faculty’s Research Ethics Committee (REC) analyzed this project (Ethical Appreciation Presentation Certificate—CAAE 27003419.1.0000.8093) and approved it under the 28 May 2020 opinion 4.055.318.

For retrospective data collection, the REC was requested to waive the Free and Informed Consent Form. Authorization was requested to release patients’ medical records to the Hospital Regional da Ceilandia (HRC) file, according to the list of patients obtained in the adult ICU admission record book for the period studied.

## 3. Results

Regarding medical records, we reviewed 844 prescriptions with a mean record of 284.9 pharmacological units per patient, which accounted for 14,248 medication doses, i.e., 14,248 opportunities for errors. The error rate was 108%, exceeding 100 percentage points, as there was more than one error for some pharmacological units. According to Table 1, the error type 4 rate corresponds to 53.63% of errors due to medication doses and 50.96% of the total observed errors, followed by the type 6, 7, and 2 error rates, respectively.

Table 2 shows the medication error (ME) type analyzed per hospitalization and day of hospitalization, demonstrated by their mean and standard deviation. The omission error presented the highest mean in both analyses, according to hospitalization and day of hospitalization, followed by the time, prescription, and scheduling errors.

Table 3 shows the characteristics of the study variables included in the regression model, which shows a mean hospitalization cost (MHC) of BRL 8169.78 (about USD $1676), a male predominance, 59.32 ± 19 years, and 16.7 ± 16 days of hospitalization.

Table 4 presents the results of the regression models. Hospitalization cost correlated positively and significantly with ME type 2 (scheduling error) and 7 (prescription error) in the simple and multivariate models, suggesting that, as scheduling or prescription errors increase, so will hospitalization costs. For example, per each additional type 2 error unit, the hospitalization cost would increase by BRL 121.92 (about USD $25.00) (95% CI 43.09; 200.74), while per each additional type 7 error unit, the cost would increase by BRL 63.51 (about USD $13.00) (95% CI 29.93; 97.09), taking into account the multivariate model (with controls). Error types 4 (omission errors) and 6 (time errors) were only statistically significant in the simple analysis, in which confounding variables did not control them. In these cases, an increase in the ME amount also positively correlated with increased hospitalization costs.

## 4. Discussion

The diversity of contexts creates challenges for comparing studies, increasing the difficulty of producing reproducible scientific evidence to support clinical decision-making and health service management [13,15,16,17]. This study makes a positive contribution in this direction by presenting the scenario of a public hospital and highlighting the importance of local data, given the diversity of contexts and practices between hospitals in the country and worldwide.

The study and validation of care indicators related to the medication process face difficulties and limitations, such as a scarcity of notifications, lack of records, low medication error (ME) detection when no damage is caused, and the complexity of the stages involved in the medication chain. Despite this, examining these indicators is essential and relevant since it subsidizes coherent decision-making, even when the minimum scores for validation are unreached [23,24,25,26]. The present study also had these difficulties and limitations as the data were collected through medical records and, as such, an estimated probability of failure to record information.

Our findings draw attention to the total error rate (108%) and percentage of error per administered medication dose, with emphasis on errors of omission (53.63%, error type 4), time (25.54%, error type 6), and prescription (11.32%, error type 7), and also on prescription and scheduling (error type 2) error frequencies when analyzing the error per day of hospitalization and increase in hospitalization costs. Several authors have reported a high error rate, ranging from 43% to 118% [3,4,5,6,27,28]. When analyzing the available data on error rate and opportunity for error, some authors found more than one error per patient, corroborating our findings [3,4]. Assunção-Costa et al. [5] pointed to a high error rate associated with medication in Latin America, with a 32% median error and high variability in the frequencies described.

Most ME studies indicate that prescription, omission, and scheduling errors are the most recurrent error types. Studies investigating errors related to the medication process have observed a higher prevalence of omission errors [14,21,27,28,29,30], followed by prescription errors. Varizi et al. [31] concluded that prescription errors stand out compared to administration and prescription errors, and these findings have been reinforced by the work of other researchers [3,4,32,33,34]. Suclupe et al. [3] point out a 67.6% prescription error rate with an average of 7 errors per patient, while another research group found a 72% prescription-related error rate with an average of 13.1 errors per patient.

Karthikeyan et al. [33] reported prescription error frequencies ranging from 7.1 to 68.2%, as did Mekonnen et al.’s [34] study, in which the average of prescription errors was 57.4% with a range between 22.8% and 77.8%. On the other hand, Pimentel et al. [20] found a 29.7% incidence of scheduling errors and a 76.4% incidence of prescription errors in their study, whereas Paulino et al. [35] reported a 27% incidence of dose omission errors.

Röhsig et al.’s [11] five-year integrative review on the cost of errors in the medication chain revealed that 94.7% of the authors analyzed the medication prescription stage at some point in their studies and that the average avoidable cost of this type of error corresponds to more than BRL 3 billion/year. Ranchon et al. [36], when studying errors related to cancer patients undergoing chemotherapy treatment, reported that the 436 errors identified could generate 216 additional days of hospitalization and an avoidable cost/year of approximately BRL 624,186.38.

Our research results suggest that some ME types impact hospitalization costs more than others, especially scheduling and prescription errors. Namely, for each additional type 2 error (scheduling error), hospitalization costs would increase by BRL 121.92, while for type 7 error (prescription error), hospitalization costs would increase by BRL 63.51. Other studies have confirmed this finding, verifying an increase in the average hospitalization cost associated with ME [15,35,37]. Interestingly, when analyzing these authors’ results, an increase in hospitalization costs correlated with the use of the ICU service and an average hospital stay longer than seven days, probably because these two factors lead to an increase in the use of technologies, more complex care, and increased hospital expenses.

Another point to highlight, besides the higher costs due to error, is the loss of quality of life. Kirwan et al. [37] calculated the costs and potential consequences associated with ME at hospital discharge using economic cost–utility analysis in health and found an increase in expenses when evaluating the proportion of error, as well as a reduction in QALYs (Quality-Adjusted Life Years).

It is crucial to strengthen the teams’ permanent education programs to raise awareness about the importance of registering medical records and reporting notifications about errors related to the medication process, as well as other strategies reported in some studies, such as the computerization of medication dispensing, double-checking, unitized (single) dose, patient and family empowerment, the prescriber’s knowledge, medication reconciliation led by the pharmacist, among others [20,23,25,26].

Against this background, it is imperative to study the ME-related costs for their impact on health institutions, patients and their families, and society as a whole, as their minimization directly affects the health economy by reducing these expenses that generate a potential benefit for interventions aiming to address these types of errors [15,17,24,25,35]. Accordingly, it is paramount to identify the factors that cause medication errors, their damage, and the costs passed on in this process.

Among some of this study’s limitations are possible data collection problems due to the failure to register some electronic medical records and the fact that the costs are based on the transfer values of the Ministry of Health and do not necessarily reflect the hospital cost. Regardless, these findings are of great importance as they provide insights into the workflows and processes within ICUs and reflect how errors in practice negatively affect the financial health of health institutions.

## 5. Conclusions

Medication errors can potentially cause harm to patients, prolong hospitalization periods, affect treatment, increase health costs, and unsettle the global health system. Hence, it is imperative to implement strategies that identify and minimize predisposing factors that increase the likelihood of these errors.

Our study highlights the importance of knowing, identifying errors, and relating them to the cost of hospital admissions. Seven types of medication errors were identified, with a higher prevalence for errors of omission, time, and prescription (in order of severity). Our results also suggest that two types of medication errors were associated with increased hospital admission costs for an adult ICU (scheduling and prescription errors).

Given these findings, using technology such as barcodes for drugs and patients, smart infusion pumps for intravenous delivery, single-use drug packaging, and packaging design resources could reduce these drug administration errors by helping institutions to heed the nine rights of drug administration: right patient, right drug, right route, right time, right dose, right documentation, right action, right way, and right answer.

## Figures and Tables

**Table 1 ijerph-20-06788-t001:** Type of errors identified and their frequency in the Adult Intensive Care Unit (ICU) of a general hospital in the Federal District, DF, Brazil, 2019.

Type of Error	Frequency of Medical Records with Errors (14,372 Medication Doses)—%	Total Errors Identified in Each Dose (N = 15,123)	%
Error type 1	0.35	51	0.34
Error type 2	6.75	970	6.41
Error type 3	1.26	181	1.20
Error type 4	53.63	7707	50.96
Error type 5	1.15	165	1.09
Error type 6	25.54	3671	24.27
Error type 7	11.32	1627	10.76

Note: Statistics were calculated based on 50 patients. The total number of errors exceeds the number of pharmacological units analyzed, as there was more than one error for some pharmacological units. Legend: Error type 1 = dosage error; Error type 2 = scheduling error; Error type 3 = administration route error; Error type 4 = omission error; Error type 5 = dispensing error; Error type 6 = time error; Error type 7 = prescription error.

**Table 2 ijerph-20-06788-t002:** The mean number of medication errors per hospitalization and per day in the Adult Intensive Care Unit (ICU) of a general hospital in the Federal District, DF, Brazil, 2019.

Medication Error Type	Per Hospitalization				Per Day of Hospitalization			
	Mean	Standard Deviation	Minimum	Maximum	Mean	Standard Deviation	Minimum	Maximum
Error type 1	1.16	2.61	0	13	0.60	2.15	0	13.0
Error type 2	22.05	34.32	0	161	1.49	1.89	0	7.5
Error type 3	4.11	10.36	0	61	1.01	4.62	0	30.5
Error type 4	175.16	178.82	0	657	13.25	13.47	0	57.0
Error type 5	3.75	8.24	0	34	1.12	3.50	0	17.0
Error type 6	83.43	100.32	0	402	5.03	5.41	0	28.9
Error type 7	36.98	80.25	0	434	1.93	3.40	0	19.6

Note: Statistics were calculated based on 50 patients. Legend: Error type 1 = dosage error; Error type 2 = scheduling error; Error type 3 = administration route error; Error type 4 = omission error; Error type 5 = dispensing error; Error type 6 = time error; Error type 7 = prescription error.

**Table 3 ijerph-20-06788-t003:** Descriptive analysis of the variables included in the regression models in the Adult Intensive Care Unit (ICU) of a general hospital in the Federal District, DF, Brazil, 2019.

Variable	N° Observations	Mean/ Frequency	Standard Deviation	Minimum	Maximum
Hospitalization cost (MHC) *	44	8169.78	11,236.28	342.54	58,254.71
Days of hospitalization	44	16.66	16.07	1.00	70.00
Age	44	59.32	19	18	97
Biological sex (=1, if female)	44	9 (21%)	-	0	1
Death (=1, if death)	44	27 (61%)	-	0	1
Hypertension	44	30 (68%)	-	0	1
Diabetes Mellitus	44	13 (29%)	-	0	1
Alcoholic	44	30%	-	0	1
Smoker	44	32%	-	0	1

Note *: values expressed in reais (current currency, BRL) for hospitalization costs. (USD $1.00 = BRL 4.87).

**Table 4 ijerph-20-06788-t004:** Results of the simple (without controls) and multivariate (with controls) regression models regarding the association between hospitalization costs and medication error types in the Adult Intensive Care Unit (ICU) of a general hospital in the Federal District, DF, Brazil, 2019.

Medication Error Type		Coefficient	Standard Error	*p*-Value	Confidence Interval (95%)	R^2^
Error type 1	without control	−18.63	413.83	0.826	(−851.62; 862.53)	1.1%
	with control	77.86	312.26	0.725	(−525.45; 724.24)	56.27%
Error type 2	without control	224.17	36.82	<0.01 *	(149.86; 298.48)	46.9%
	with control	125.28	27.061	<0.01 *	(32.12; 213.83)	69.2%
Error type 3	without control	−88.03	166.85	0.601	(−424.75; 248.71)	0.7%
	with control	−112.13	112.65	0.456	(−342.73; 124.97)	55.42%
Error type 4	without control	27.75	8.70	0.003 *	(10.21; 45.30)	19.5%
	with control	7.32	8.23	0.493	(−7.98; 22.69)	53.33%
Error type 5	without control	−57.65	210.21	0.785	(−481.86; 366.55)	0.02%
	with control	33.28	152.23	0.777	(−232.33; 341.75)	56.25%
Error type 6	without control	34.42	16.45	0.042 *	(1.23;67.61)	9.4%
	with control	−0.53	15.24	0.875	(−26.03; 29.12)	55.43%
Error type 7	without control	97.78	15.47	<0.001 *	(66.58; 128.99)	48.8%
	with control	67.29	17.78	<0.001 *	(26.75; 99.89)	68.23%

Note: “Without control” refers to the simple linear model, with only the type of error as an independent variable. “With control” refers to the multivariate linear model, which included the following independent variables: (i) the type of medication error; (ii) days of hospitalization; (iii) age of the patient at the time of hospitalization; (iv) sex of the patient; (v) whether the patient had hypertension; (vi) whether the patient had diabetes mellitus; (vii) whether the patient was an alcoholic at the time of hospitalization; and (viii) whether the patient was a smoker at the time of hospitalization.. * *p* < 0.05. BRL 1.00 = USD $0.20.

## Data Availability

The article’s tables contain the research data.

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
