# Peer review of "Determining Medication Errors in an Adult Intensive Care Unit"

_ijerph, 2023, doi:10.3390/ijerph20186788_

Round 1

Reviewer 1 Report

Keywords: Instead of Medical error it would be better to use Medication error

Line 45: Would not use new term such as Medication-related errors - it would be better to stick with one term throughout the paper

Line 46: Why not 750000 EUR instead of 0.75?

Line 58: the not t

------line 158: is or was a smoker

Line 156-158: Could you please add one sentence why you made notes on this specific states: diabetes, alcoholic, smoker, high blood pressure?ž

Overall: it would be better for all readers to calculate each amount in dollars as well.

Under Table 4: Please put the Note in English.

Conclusion: Maybe not to repeat the percentages, better to emphasize the need for better medication error reporting among health care professionals and to emphasize the type of predominant errors and possible preventable methods.

Author Response

We appreciate the suggestions made and addressed each point raised by the reviewers. We believe that these suggestions increased the overall quality of the submitted manuscript.

Therefore, we are resubmitting our paper entitled " Determining medication errors in an adult intensive care unit.” 

All authors are aware of the resubmission and agree with the responses to the reviewers provided below.

Reviewer #1

Keywords: Instead of Medical error it would be better to use Medication error

Thank you for suggesting this; we have incorporated it into the text.

Line 45: Would not use new term such as Medication-related errors - it would be better to stick with one term throughout the paper

Thank you for the suggestion; we adopted the term “Medication error” throughout the text (“Medical error”/“Medication-related errors” to “Medication error”).

Line 46: Why not 750000 EUR instead of 0.75?

Thank you very much for the suggestion, but we would prefer the values to stay in the same unit.

Line 58: the not t

Sorry, we didn't understand the suggestion

------line 158: is or was a smoker

Patient was a smoker at the time of hospitalization

Line 156-158: Could you please add one sentence why you made notes on this specific states: diabetes, alcoholic, smoker, high blood pressure?ž

Thank you very much for this suggestion; we added this phrase to the end of 2.6. Data processing and analysis' last paragraph:

'These independent variables were chosen as they are known to impact the length of the hospitalization stay.'

Overall: it would be better for all readers to calculate each amount in dollars as well.

Thank you very much for this suggestion. We fully agree and have put the conversion values in dollars in the text.

Under Table 4: Please put the Note in English.

Forgive our inattention. The note has been rewritten in English.

Conclusion: Maybe not to repeat the percentages, better to emphasize the need for better medication error reporting among health care professionals and to emphasize the type of predominant errors and possible preventable methods.

Thank you very much for your contribution. We agree and have rewritten the paragraph to:

Medication errors can potentially cause harm to the patient, prolong the hospitalization period, affect treatment, increase health costs, and unsettle the global health system. Hence, it is imperative to implement strategies that identify and minimize predisposing factors that increase the likelihood of these errors.

Our study highlights the importance of knowing, identifying errors, and relating them to the cost of hospital admissions. Seven types of medication errors were identified, with a higher prevalence for errors of omission, time, and prescription (in order of preponderance). Our results also suggest that two types of medication errors were associated with increased hospital admission costs to an adult ICU (scheduling and prescription errors).

Given these findings, using technology such as bar codes for drugs and patients, smart infusion pumps for intravenous delivery, single-use drug packaging, and packaging design resources could reduce these drug administration errors by helping to heed the nine rights of drug administration: right patient, right drug, right route, right time, right dose, right documentation, right action, right way, and right answer

Reviewer 2 Report

I recommend changes to the executive summary. INTRODUCTION - AIM OF THE STUDY - METHODOLOGY - RESULTS - CONCLUSIONS.

No chapter - AIM OF THE STUDY. Please extract.

Conclusions are duplication of results. I recommend writing out the conclusions from the points as the most important evidence arising from the study.

I suggest adding a chapter on IMPLICATIONS - which will be a "signpost" for medical institutions.

Author Response

We appreciate the suggestions made and addressed each point raised by the reviewers. We believe that these suggestions increased the overall quality of the submitted manuscript.

Therefore, we are resubmitting our paper entitled " Determining medication errors in an adult intensive care unit.” 

All authors are aware of the resubmission and agree with the responses to the reviewers provided below.

Reviewer #2

I recommend changes to the executive summary. INTRODUCTION - AIM OF THE STUDY - METHODOLOGY - RESULTS - CONCLUSIONS.

Thank you very much for this suggestion; we have executed the change, although the Journal template strongly suggested the previous one.

No chapter - AIM OF THE STUDY. Please extract.

Thank you very much for this suggestion, but we follow the chapters described in the template for submission.

Conclusions are duplication of results. I recommend writing out the conclusions from the points as the most important evidence arising from the study.

Thank you very much for your contribution. We agree and have rewritten the paragraph to:

Medication errors can potentially cause harm to the patient, prolong the hospitalization period, affect treatment, increase health costs, and unsettle the global health system. Hence, it is imperative to implement strategies that identify and minimize predisposing factors that increase the likelihood of these errors.

Our study highlights the importance of knowing, identifying errors, and relating them to the cost of hospital admissions. Seven types of medication errors were identified, with a higher prevalence for errors of omission, time, and prescription (in order of preponderance). Our results also suggest that two types of medication errors were associated with increased hospital admission costs to an adult ICU (scheduling and prescription errors).

Given these findings, using technology such as bar codes for drugs and patients, smart infusion pumps for intravenous delivery, single-use drug packaging, and packaging design resources could reduce these drug administration errors by helping to heed the nine rights of drug administration: right patient, right drug, right route, right time, right dose, right documentation, right action, right way, and right answer.

I suggest adding a chapter on IMPLICATIONS - which will be a "signpost" for medical institutions.

Thank you very much for this suggestion, we write this on conclusion

Given these findings, using technology such as bar codes for drugs and patients, smart infusion pumps for intravenous delivery, single-use drug packaging, and packaging design resources could reduce these drug administration errors by helping to heed the nine rights of drug administration: right patient, right drug, right route, right time, right dose, right documentation, right action, right way, and right answer.

Reviewer 3 Report

I have nit any comments!

Author Response

Thank you for you consideration

Reviewer 4 Report

English is very poor it need urgent fixation

It should be re-written as contents not very clear

1. Introduction

Line 29 and 31: Abbreviation is developed twice: . Medication errors (ME), Medication Error (ME)

Its confusing. Please write as Medication Error (ME) and Medication errors (MEs) or simply MEs for Medication errors

Line 40: Research addressing ME costs is still scarce. Please improve the English.

STUDY GAP AND OBJECTIVE is not very clear in the introduction

2.  Method

a. no sample calculation and sampling

b, author mention in the introduction and abstract about the cost; How the cost was calculated ?

c. sub heading need modification like Site/Locus, Study Type and Ethical aspects

3. Results

a. No demographics there.

c. Table 1. Error type 1,2....................... its confusing. mention the name of error

Discussion

not illustrative and informative.

Conclusions

it should not contain results

make a summary and suggestions

Need major correction

Author Response

We appreciate the suggestions made and addressed each point raised by the reviewers. We believe that these suggestions increased the overall quality of the submitted manuscript.

Therefore, we are resubmitting our paper entitled " Determining medication errors in an adult intensive care unit.” 

All authors are aware of the resubmission and agree with the responses to the reviewers provided below.

Reviewer #4

  1. Introduction

Line 29 and 31: Abbreviation is developed twice: . Medication errors (ME), Medication Error (ME)

Thank you very much for this suggestion; we have standardized terms for Medication Error (ME) and Medication errors (MEs).

Its confusing. Please write as Medication Error (ME) and Medication errors (MEs) or simply MEs for Medication error

Thank you very much for this suggestion; we have standardized terms for Medication Error (ME) and Medication errors (MEs).

Line 40: Research addressing ME costs is still scarce. Please improve the English.

We have altered the phrase to:

‘Research addressing the costs of ME is still scarce despite issues related to patient safety causing a significant economic and health impact.’

STUDY GAP AND OBJECTIVE is not very clear in the introduction

As stated throughout our manuscript, few studies address the costs of medication error (ME) and its economic and health impact. Therefore, our objective was to fill this gap by identifying the number of medication errors and verifying whether this number was associated with increased hospitalization costs for patients in an Intensive Care Unit (ICU).

  1. Method
  2. no sample calculation and sampling

Thank you very much for your suggestion. We have put it in point 2.2 population.

b, author mention in the introduction and abstract about the cost; How the cost was calculated ?

As pointed out in the methods, we considered healthcare costs used by the patients during the inpatient care. We used a top-down approach to measure these costs, based on values reimbursed by the Ministry of Health to health facilities.

  1. sub heading need modification like Site/Locus, Study Type and Ethical aspects

Thank you very much for your suggestion. We put the new subtitules on the text

  1. Results
  2. No demographics there.

These data are shown in table 3.

  1. Table 1. Error type 1,2....................... its confusing. mention the name of error

We described them on the tables’ footnote.

Discussion

not illustrative and informative.

We need help determining which parts are not illustrative or informative. We discussed medication errors and their impact on hospitalization costs. Is there another way in which this impact can be made more clear? Thank you for your time and consideration.

Conclusions

it should not contain results make a summary and suggestions

 Thank you very much for your contribution. We agree and have rewritten the paragraph to:

Medication errors can potentially cause harm to the patient, prolong the hospitalization period, affect treatment, increase health costs, and unsettle the global health system. Hence, it is imperative to implement strategies that identify and minimize predisposing factors that increase the likelihood of these errors.

Our study highlights the importance of knowing, identifying errors, and relating them to the cost of hospital admissions. Seven types of medication errors were identified, with a higher prevalence for errors of omission, time, and prescription (in order of preponderance). Our results also suggest that two types of medication errors were associated with increased hospital admission costs to an adult ICU (scheduling and prescription errors).

Given these findings, using technology such as bar codes for drugs and patients, smart infusion pumps for intravenous delivery, single-use drug packaging, and packaging design resources could reduce these drug administration errors by helping to heed the nine rights of drug administration: right patient, right drug, right route, right time, right dose, right documentation, right action, right way, and right answer.

Round 2

Reviewer 4 Report

The presence of few studies is not a research gap. Please mention it.

2.3. Location

Please write the study setting/s

Moderate improvement.

Author Response

Dear reviewer,

We are grateful for your comments.

1- “The presence of few studies is not a research gap. Please mention it.”

Yes, we agree that few studies are not a research gap. Nevertheless, given the importance of its health and economic impact, more ME cost research in different contexts will help improve policies that minimize its effect. 

In our manuscript, we expressed this counterpoint that ME costs inherently carry to patients’ health and both the public and private economy in the excerpt:

“Research addressing the costs of ME is still scarce despite issues related to patient safety causing a significant economic and health impact.”

2- 

“2.3. Location

 Please write the study setting/s”

Thank you for your comment. We believe that we have placed the study setting fragmented throughout the methodology’s subtopics summarized below:

2.3. Location: The research was conducted in a public hospital institution part of the Unified Health System (SUS) located in the Federal District (DF). This hospital is located in the Federal District's largest and most populous administrative region with roughly 500,000 inhabitants, corresponding to the 13th most populated municipality in the country. The selected adult ICU has ten beds, one of which is surgical, with a 14 days-stay mean and a 95% occupation rate. The factors that endorsed this unit selection were the medication volume, the medication process's complexity (calculation of doses, diversity of injectable medications, and variety of infusions), the users' clinical conditions, and the types of medications used.

2.4. Selection criteria: Patients over 18 years old with medical records that included identification information and clinical evolution during the hospitalization period in the unit, which would allow their evaluation, were included. 

2.5. Data collection: Initially, patients hospitalized during the study period were determined through the ICU admission book, and their medical records were located and accessed in the institution's electronic system. Based on this access, clinical summaries were analyzed to extract demographic, epidemiological, and clinical data. The medication information from all medical prescriptions referring to the patient's hospitalization period and the nursing record of medications' scheduling and administration were likewise collected, …” 

We also further described it in the manuscript.

Round 3

Reviewer 4 Report

.